# Application and Advantages of the Trans-Unco-Discal (TUD) Approach for Cervical Spondylotic Myelopathy and Radiculopathy: Classification and Modification of Surgical Technique Based on the Location of Spinal Cord and/or Nerve Root Compression

**DOI:** 10.3390/jcm13092666

**Published:** 2024-05-02

**Authors:** Misao Nishikawa, Kentaro Naito, Masaki Yoshimura, Toru Yamagata, Keiichi Iseda, Mitsuhiro Hara, Hiromichi Ikuno, Kenji Ohata, Takeo Goto

**Affiliations:** 1Department of Neurosurgery, Moriguchi-Ikuno Memorial Hospital, 6-17-33 Satanakamachi, Moriguchi City 570-0002, Osaka, Japan; punchdrunkard830@yahoo.co.jp (T.Y.); keiiseda@pop21.odn.ne.jp (K.I.); 2Department of Neurosurgery, Osaka Metropolitan University Graduate School of Medicine, 1-4-3 Asahimachi, Abeno-ku, Osaka 545-8595, Osaka, Japan; 7110ken622@omu.ac.jp (K.N.); gotot@omu.ac.jp (T.G.); 3Department of Neuropathology, Yao Tokusyukai General Hospital, 1-17, Wakakusacho, Yao City 581-0011, Osaka, Japan; masaking33@gmail.com; 4Department of Neurology, Moriguchi-Ikuno Memorial Hospital, 6-17-33 Satanakamachi, Moriguchi City 570-0002, Osaka, Japan; mhara-senri@kra.biglobe.ne.jp; 5Department of Neuroradiology, Moriguchi-Ikuno Memorial Hospital, 6-17-33 Satanakamachi, Moriguchi City 570-0002, Osaka, Japan; hi@koudoukai.jp; 6Department of Neurosurgery, Naniwa-Ikuno Hospital, 1-10-3 Daikoku Naniwa-ku, Osaka 556-0014, Osaka, Japan; kohata@omu.ac.jp

**Keywords:** trans-unco-discal approach, anterior cervical decompression, cervical spondylotic myelopathy, cervical spondylotic radiculopathy, operative technique

## Abstract

**Purpose:** We assess the application and advantages of modifying the trans-unco-discal (TUD) approach which we underwent for cervical myelo-radiculopathy. We present the surgical techniques of the modified TUD approach. **Materials and Methods:** The material was 180 cases where anterior cervical decompression (ACD) was performed by the modified TUD approach. We classified the material into four groups based on the location of the nerve root and/or spinal cord compression: I, compression of the root at intervertebral foramen (IVF); II, compression of the posterior margin of the vertebral body; III, compression of the IVF and posterior margin of the vertebral body; IV, compression of the bilateral IVF and posterior margin of the vertebral body. We applied the modified TUD approach to these four types. We present the surgical procedures and techniques for the modified TUD approach. The Japanese orthopedic association (JOA) score and neuroradiological alignment were examined. **Results:** The improvement rate of the JOA score was 78.4% at 6 months post-surgery and 77.5% in the most recent examinations. By the modified TUD approach, compressive lesions of the spinal cord and/or nerve roots were removed, and good alignment was acquired and sustained. **Conclusions:** ACD by the modified TUD approach safely achieved appropriate decompression for the spinal cord and/or nerve roots, and the patients had a high improvement rate and good alignment. Complications were less common than with other surgical procedures. If the TUD approach and endoscopic approaches can be combined, their application to new area is anticipated.

## 1. Introduction

The trans-unco-discal (TUD) approach, also known as the anterior lateral combined approach, was introduced by Hakuba A. in 1976 in the Journal of Neurosurgery (JNS) for anterior decompression of the cervical spine. It involves the removal of the uncinate process on the anterior side of the vertebral body to reach the lesion on the posterior side [1]. In the TUD approach, under direct vision using an operative microscope, we can safely and completely remove extensive lateral spurs without exposing the vertebral artery in its canal. There is less risk of injury to the spinal cord with the TUD approach than with a transdiscal approach. The extensive removal of the posterolateral corner and transverse ridge of the vertebral body allows sufficient space for the nerve roots and spinal cord, and there is little danger of encroachment on the intervertebral foramen and spinal canal. The usefulness and safety of the TUD approach have been confirmed [2]. Since Hakuba reported the TUD approach as a method of anterior foraminotomy of cervical spondylosis, various modifications have been reported [3,4,5]. The original TUD approach has also been modified by the following surgeons [6,7].

Anterior cervical decompression (ACD) and anterior cervical discectomy and fusion (ACDF) are common procedures in the field of spinal surgery, but the surgical methods vary. Currently, the assessment of compression sites of the spinal cord and nerve roots, as well as the degree of decompression achieved, is left to the discretion of individual surgeons. We think that it is important to perform more appropriate decompression and less invasive surgery. Therefore, it is necessary to classify compression sites of the spinal cord and/or nerve roots to identify the areas that require bone removal and decompression pre-operatively, as well as to assess the achievement and degree of decompression. The TUD approach we use can address various pathologies and compression sites. Furthermore, the TUD approach offers a safer procedure to confirm the achievement and degree of decompression.

In this study, we review the reports regarding ACD, introduce the surgical procedure and technique of the TUD approach, and examine the advantages of the TUD approach. Furthermore, we classified our cases into four groups based on the location of the compressive lesion at the spinal cord and/or nerve roots and the neurological symptoms and signs. We modified and applied the TUD approach to each group. We present the surgical techniques for the modified TUD approach for each group. We discuss the advantages and utility of the modified TUD approach for cervical spondylotic myelopathy and radiculopathy.

## 2. Materials and Methods

### 2.1. Materials

We performed the modified TUD approach on more than 2000 cases. Out of these cases, this study includes 180 cases (250 levels) cases who underwent surgery by a single surgical team using the same fixation method at Moriguchi-Ikuno Memorial Hospital (MIMH), Yao Tokusyukai General Hospital (YTGH), and Osaka Metropolitan University Hospital (OMUH) from April 2017 to March 2022.

### 2.2. Classification (Figure 1)

The compression sites of the cervical cord and nerve roots were classified into four groups.

Type I (Figure 1A,B): compression of the nerve roots in the intervertebral foramen and radiculopathy.Type II (Figure 1C,D): compression of the spinal cord in the posterior direction due to disc hernia, osteophytes, ossification of the posterior longitudinal ligament, and myelopathy.Type III (Figure 1E,F): compression of the nerve roots and spinal cord posteriorly from the intervertebral foramen, and myelopathy and/or radiculopathy.Type IV (Figure 1G,H): in addition to Type III, compression of the spinal cord and nerve roots on the opposite side of the intervertebral foramen, and myelopathy and/or bilateral radiculopathy.

**Figure 1 jcm-13-02666-f001:**
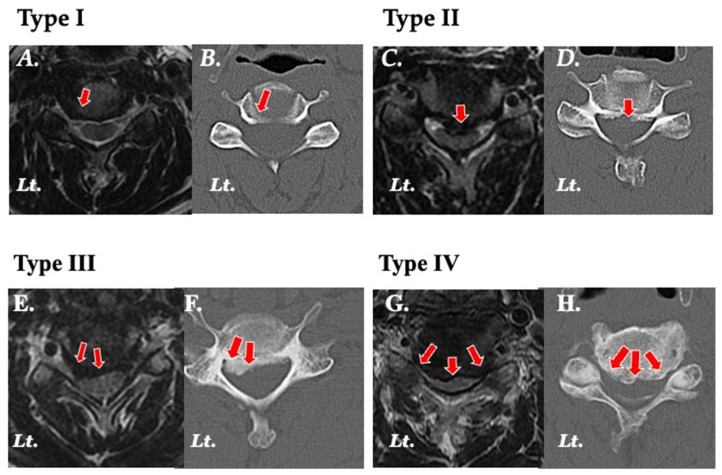
Classification based on the location of the compression. Magnetic resonance (MR) axial images (left) and two-dimensional computed tomography (2D-CT) images (right) show the location of the lesion (red arrows) in each type. (**A**,**B**) Type I: showing an osteophyte and disc hernia and compression for the nerve root in the left intervertebral foramen. (**C**,**D**) Type II: showing a mild osteophyte median dorsal vertebral body and severe disc hernia at the midline and compression for the spinal cord. (**E**,**F**) Type III: showing osteophyte in the intervertebral foramen and dorsal vertebral body and severe compression of the nerve root and spinal cord. (**G**,**H**) Type IV: showing an osteophyte in the bilateral intervertebral foramen and dorsal vertebral body and severe compression for the nerve root and spinal cord.

### 2.3. Special Equipment (Figure 2)

In terms of surgical instruments, we use a specially designed spreader, sharp curette, and Kerrison rongeur (Figure 2A–C). The spreader’s tips are tapered and grooved, facilitating easy insertion into the narrow intervertebral space without damaging the endplate of the vertebral body (Figure 2A). The curette and Kerrison rongeur also have a spreader tip to minimize tissue disruption (Figure 2B,C). We use Brown–Aesculap Ellan drill, which has an integrated motor for smooth and stable rotation (Figure 2D).

**Figure 2 jcm-13-02666-f002:**
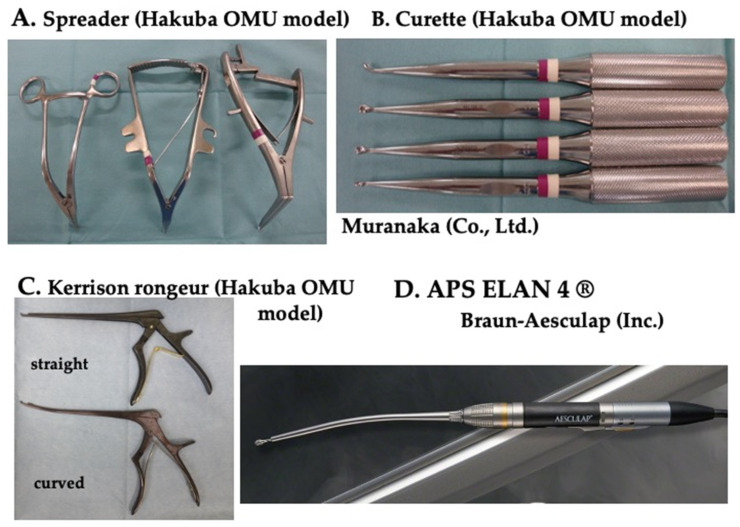
Special equipment. (**A**) Spreader. (**B**) Curette. (**C**) Kerrison rongeur. (**D**) High-speed ELAN drill (Brown–Aesculap, Melsungen, Germany).

### 2.4. Surgical Procedure of the Original TUD Approach (Figure 3)

#### 2.4.1. Skin Incision, Exposure of the Ventral Vertebral Body, and Burr Hole (Figure 3A–D)

A collar incision, according to the operative level along the wrinkle, is made and the approach to the ventral surface of the vertebral body is via the cervical triangle (Figure 3A). In the case of the left-sided approach, the recurrent laryngeal nerve runs along the back of the esophagus, so care is taken not to detach the posterior aspect of the esophagus while reaching for the anterior aspect of the vertebral body. Additionally, when exposing the esophagus along the vertebral body, slight traction of the esophagus along the deep cervical fascia is applied. Conversely, in the approach from the right side, the recurrent laryngeal nerve runs along the side of the esophagus, so effort is made to minimize traction on the esophagus. Sometimes, surgery is performed using a thin cotton sheath as protection. The longus colli muscle is divided transversely, and the stumps are retracted with a retract so that the lateral osteophyte is exposed (Figure 3B). A burr hole is made at the medial side of the base of the uncinate process (Figure 3C). The posterior lateral osteophyte is located near the vertebral artery (Figure 3D) [1,2,6,7].

**Figure 3 jcm-13-02666-f003:**
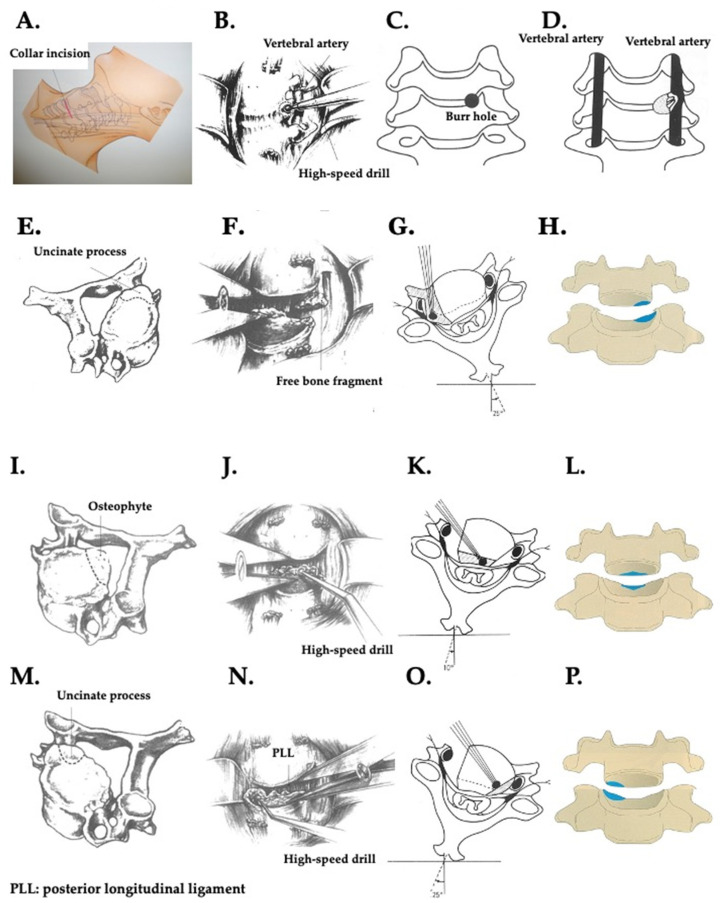
Sketches and shema of the trans-unco-discal (TUD) approach (approach via the left side approach). (**A**) Collar incision according to the operative level along the wrinkle. (**B**) The prevertebral space is exposed. (**C**) The position of the burr hole at the medial side of the base of the uncinate process. (**D**) The posterior lateral osteophyte located near the vertebral artery. (**E**) Removal of the disc at the lateral osteophyte (broken line). (**F**) After the removal of the disc hernia at the lateral and posterior sites, the surgeon’s side is manually retracted and a vertebral spreader is inserted in the opposite side to open the vertebral space and then retracted. (**G**) The operating table is rotated approximately 25° toward the surgeon. The angle of the drill and the area were removed by uncinectomy and foraminotomy (black shadow area). (**H**) The posterior lateral corner of the vertebral body (blue area). (**I**) Removal of the portion of the dorsal osteophyte of the vertebral body (broken line). (**J**) The surgeon’s side is manually retracted and a vertebral spreader is inserted in the opposite side to open the vertebral space and then retracted. (**K**) The operating table is rotated an additional 10° away from the surgeon and the dorsal vertebral body osteophyte is removed using a high-speed drill. The angle of the drill and dissected area (black shadow area). (**L**) Dissected area (blue area). (**M**) The broken line indicates the plane of resection of the opposite uncinate process and lateral osteophyte (broken line). (**N**) The opposite side is manually retracted and a vertebral spreader is inserted in the surgeon’s side to open the vertebral space and then retracted. (**O**) The operating table is rotated 25° from the horizontal plane away from the surgeon. The angle of the drill and the dissected area (black shadow area). (**P**) The posterior lateral corner of the vertebral body (blue area).

#### 2.4.2. Unco-Foraminotomy (Figure 3E–H)

An operative microscope is then brought into the surgical field and using a high-speed drill with an angled adaptor and suction irrigation, the anterior portion of the uncinate process is removed first, followed by the removal of the lateral osteophyte (Figure 3E). After removal of the disc mass with a curette and Kerrison rongeur, a vertebral spreader is inserted (Figure 3F). To obtain direct sight of the medial wall of the uncinate process and posterolateral corner of the vertebral body through the operative microscope, the operating table is rotated approximately 25° toward the surgeon (Figure 3G). Emptying the disc space and widening it with the instrument greatly facilitates exposure of the rest of the uncinate process and the remaining portion of the lateral osteophyte at the medial and posterior side of the vertebral body. Then foraminotomy is carried out under the operative microscope with a high-speed drill and curette (Figure 3F,H). Further removal of the intraforaminal osteophyte and extruded lateral free disc fragments is performed through the space obtained by complete unco-foraminotomy (Figure 3F,H) [1,2,6,7].

#### 2.4.3. Resection of Osteophytes of Dorsal Vertebral Body (Figure 3I–L)

The operative table is rotated 10° from the horizontal plane toward the side away from the surgeon. Through the space created by the removal of the uncinate process, the posterior spur is resected with a high-speed drill and curette (Figure 3I,J) both of which are inserted in a plane parallel to the dural sac. (Figure 3J). The operating table is rotated an additional 10° away from the surgeon, and the dorsal vertebral body osteophyte is removed using a high-speed drill (Figure 3K) and the curette is inserted through the space made by resection of the osteophyte and discectomy (Figure 3L) [1,2,6,7].

#### 2.4.4. Unco-Foraminotomy of Opposite Side (Figure 3M–P)

The operating table is rotated 25° from the horizontal plane toward the side away from the surgeon, and the opposite intraforaminal osteophyte and foraminotomy are carried out with a high-speed drill (Figure 3M,O). The curette is inserted through the space made by uncinectomy and discectomy (Figure 3M). Bone removal is performed easily by way of this approach (Figure 3M,P) [1,2,6,7].

### 2.5. Application and Modification of the TUD Approach for Each Type and Surgical Techniques (Figure 4, Figure 5 and Figure 6)

#### 2.5.1. Type I: Unco-Foraminotomy Is Indicated (Figure 4)

A burr hole is drilled in the elevated portion of the uncinate process, preserving the longus colli muscle and minimizing the cut. Approximately one-quarter of Luschka’s joint is opened while the uncinate process is shaved, expanding the intervertebral foramen to decompress the nerve roots (Figure 4).

**Figure 4 jcm-13-02666-f004:**
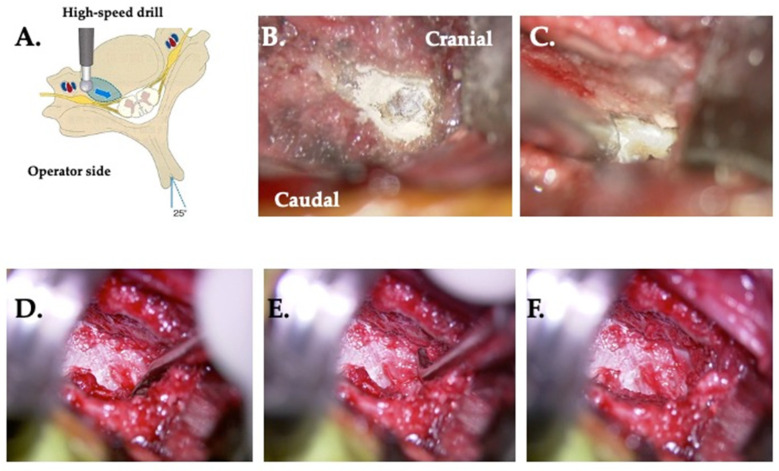
Unco-foraminotmy. (**A**) The operating table is rotated 25° toward the operator side. The area of the uncinectomy of the posterior quarter/half including the posterior lateral corner is indicated (blue area). The blue arrow shows the direction of drilling. (**B**) A burr hole is made at the medial side of the base of the uncinate process. (**C**) Drilling of the posterior lateral corner. (**D**) Resection of the caudal side osteophyte using a curette. (**E**) Resection of the cranial side osteophyte using a curette. (**F**) Perfect foraminotomy of the posterior quarter of the uncinate process and decompression of the starting point of the nerve root.

#### 2.5.2. Type II: Resection of an Osteophyte of the Dorsal Vertebral Body Is Indicated (Figure 5)

Surgery is performed inside from the posterior–lateral corner without manipulating the uncinate process. By using a spreader, the intervertebral space is opened and the surgical space is exposed by traction of both the spreader and retractor. A Kerrison rongeur is used to dissect the bone fragment and disc, and the operation time is approximately 15–20 min per side. The dissection involves lifting and excising the superficial and deep layers laterally. After confirming the dura mater, the spinal cord is then decompressed (Figure 5).

**Figure 5 jcm-13-02666-f005:**
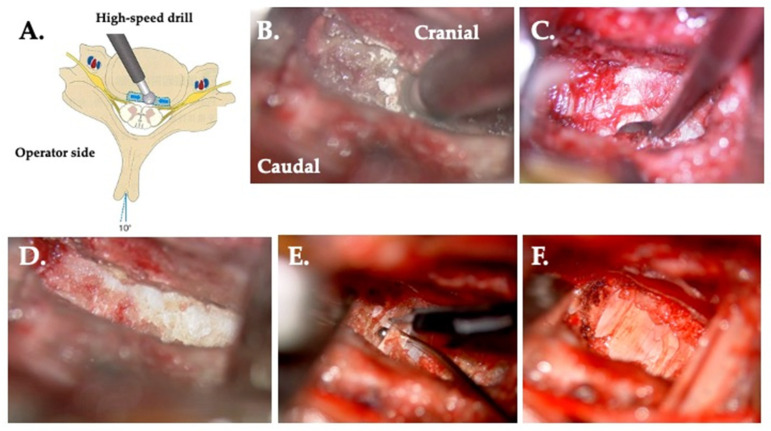
Resection of an osteophyte of the dorsal vertebral body. (**A**) The operating table is rotated 10° from the horizontal plane away from the surgeon. The area of the resection of the osteophyte at the dorsal vertebral body is indicated in blue. Blue arrows show the direction of drilling. (**B**) Drilling of the osteophyte. (**C**) Resection of the caudal osteophyte by curette. (**D**) Perfect resection of both the cranial and caudal osteophytes. (**E**) The superficial layer of longitudinal ligament is raised by a probe and cut. (**F**) Decompressed dura mater.

#### 2.5.3. Type III: Unco-Foraminotomy and Dissection of an Osteophyte of the Dorsal Vertebral Body Are Indicated (Figure 4 and Figure 5)

By removing the posterior lateral corner and opening approximately one-quarter of the posterior aspect of Luschka’s joint, decompressing of the nerve root is achieved. Resection of the osteophyte is performed posteriorly to the vertebral body. Care is taken to cut the bone spurs parallel to the spine, starting from the posterior lateral corner and moving inward. It is crucial to avoid pressing the drill against the dura mater. When removing bone from the posterior aspect of the vertebral body, the operative microscope is tilted toward the head and tail, and the drill is angled for precise removal. To remove ossification of the bone spurs and posterior ligament to reach the posterior surface of the vertebral body, approximately 5 mm of bone is removed from just before. This ensures a clear view of the posterior surface of the vertebral body, allowing for the removal of a lesion up to approximately one-third of the height of the vertebral body through a combination of the previous drill operations and the curette for ligament and bone removal (Figure 4 and Figure 5).

#### 2.5.4. Type IV: Type III plus Opposite Side Unco-Foraminotomy Are Indicated (Figure 6)

From the opposite posterior lateral corner, part of the outer uncinate process is removed, and one-quarter of the posterior aspect of Luschka’s joint is opened to decompress the nerve root. The intervertebral foramen on the opposite side is accessible, making the procedure easier with a slight tilt of the table. Surgery is performed on the outside without the posterior longitudinal ligament, and the nerve root is directly exposed, requiring careful attention to avoid damaging the nerve root, venous plexus, and vertebral artery, as well as preserving the Luschka’s joint. The removal of the posterior lateral corner stops at the existing site of the intervertebral foramen, where nerve root decompression has been achieved. From the posterior lateral corner inward, the bone spurs of the vertebral body are removed. Using a high-speed drill and curette, drilling is performed from the surgeon’s posterior lateral corner toward the midline. Decompression is considered completely when the midline is reached (Figure 6).

**Figure 6 jcm-13-02666-f006:**
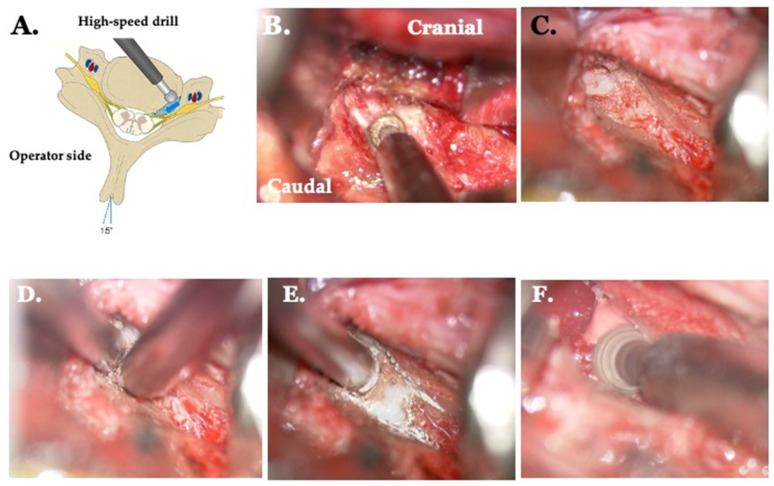
Opposite side unco-foraminotomy. (**A**) The operating table is rotated 15° from the horizontal plane away from the surgeon. The area of resection of the osteophyte at the dorsal vertebral body is indicated in blue. The blue arrow shows the direction of drilling. (**B**) Drilling of the osteophyte. (**C**) Resection of the caudal osteophyte using a curette. (**D**) Drilling of the caudal side showing the opening of the posterior lateral corner. (**E**) Drilling toward the median side. (**F**) Drilling of both the cranial and caudal sides.

#### 2.5.5. Setting of the Cage for the Interbody Fusion (Figure 7)

The cage (titanium-coated polyetheretherketone (PEEK) cage) is inserted obliquely from the anterior-lateral side into the intervertebral space. By driving stick (white asterisk), the direction and depth of the cage are adjusted, from the perspective of the surgeon, striking into the back and rotating the cage slightly. During insertion, the direction is adjusted, and the placement of the cage is confirmed to be straight in the middle using C-arm X-ray fluoroscopy.

**Figure 7 jcm-13-02666-f007:**
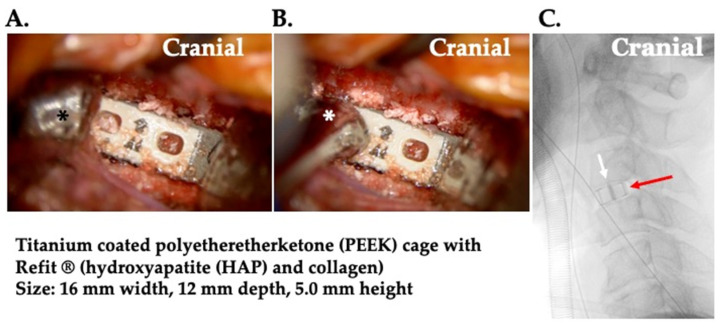
The setting of the cage. During operative photos. (**A**) the cage (titanium-coated polyetheretherketone (PEEK) cage) is inserted obliquely from the anterior–lateral side into the intervertebral space. Manual retractor (black asterisk). (**B**) By driving stick (white asterisk), the direction and depth of the cage are adjusted. (**C**) The direction is adjusted, and the placement of the cage is confirmed to be straight in the middle using C-arm X-ray fluoroscopy. It demonstrates the straight and appropriate position of the cage that the markers (white arrow) on the right and left of the center in the cage (red arrow) completely overlap.

### 2.6. Outcome Assessment

#### Assessment of Neurological and Activity of Daily Living and Japanese Orthopedic Association (JOA) Score (Table 1)

Pre- and post-operative neurological symptoms and signs were assessed and the activity of daily living was assessed using the Japanese Orthopedic Association (JOA) score at 6 months after surgery and at the most recent visit to the outpatient clinic (Table 1) [8].

Postoperatively, the neurological symptoms and signs and the improvement rate of JOA score (JOA score IR) were assessed ([post-operative points − pre-operative points]/[total points (17 points) − pre-operative point] × 100%) [8].

**Table 1 jcm-13-02666-t001:** Japanese Orthopedic Association (JOA) score.

Criterion	
Motor function	
Paraplegia	0
Paralysis	1
Upper extremity	
Fine motor function massively decreased	2
Fine motor function decreased	3
Discreet weakness in hands or proximal arm	4
Normal function	5
Motor function	
Unable to stand	0
Unable to walk	1
Lower extremity	
Need walking aid on flat floor	2
Need handrail on stairs	3
Able to walk without walking aid, but inadequate	4
Normal function	5
Sensory	
Upper extremity/lower extremity/trunk	
Sensory omission	0
Apparent sensory loss	1
Minimally sensory loss	2
Normal function	3
Bladder function	
Urinary retention	1
Sever dysfunction	2
Mild dysfunction	3
Normal function	4
Total score	0–17

### 2.7. Neuroradiological Examinations (Figure 8 and Figure 9)

The C2-7 lordotic angle (LA) was used to assess lordosis. The C2-7 sagittal vertical axis (SVA) and T1 slope were used to assess the anterior slope. The C2-7 LA indicates the angle between the lower endplate of C2 and C7. The C2-7 SVA indicates the distance from the posterosuperior corner of C7 to a vertical line from the center of the C2 vertebra. The T1 slope indicates the angle between the upper endplate of T1 and the horizontal line (Figure 8) [9,10,11].

The intervertebral space (IVS) was used to assess the spondylotic change of joints, disc degeneration, and subsidence post-operatively. The IVS was measured as the mean distance of endplates on the anterior edge of the cages in a two-dimensional computed tomography (2D-CT) scan of midline sagittal images (Figure 9).

**Figure 8 jcm-13-02666-f008:**
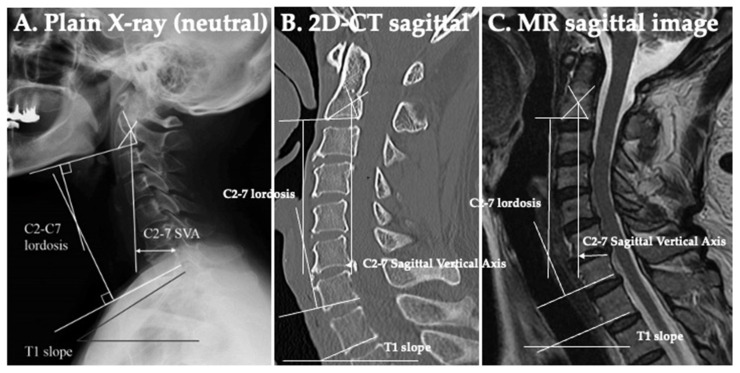
Neuroradiological examinations. The C2-7 lordotic angle (LA), C2-7 sagittal vertical axis (SVA), and T1 slope. (**A**) Plain X-ray (neutral position). (**B**) Two-dimensional computed tomography (2D-CT) scan. (**C**) T2-weighted sagittal magnetic resonance (MR) image.

**Figure 9 jcm-13-02666-f009:**
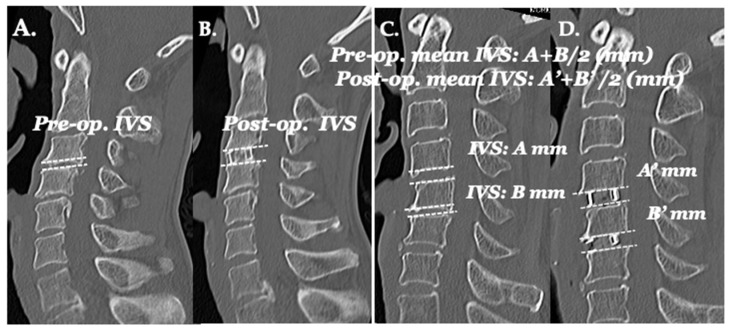
Neuroradiological examinations. Intervertebral space (IVS) is the mean distance of endplates on the anterior edge of the cages in two-dimensional computed tomography (2D-CT) scan sagittal images. (**A**) Pre-surgery. (**B**) Post-surgery. Fixation was performed in just one level to C3/4. (**C**) Pre-surgery. (**D**) Fixation was performed in two levels: C4/5 and 5/6.

### 2.8. Statistical Analyses

Statistical significance was calculated by the Stat-View 5.0 software (SAS Institute Inc., Cary, NC, USA). Surgical outcome was assessed using the Mann–Whitney test to compare the average between two groups (JOA score IR), and a paired *t*-test to compare between pre- and post-operative data (neuroradiological data). A *p*-value < 0.01 was used to determine significance.

## 3. Results

### 3.1. Classification (Table 2)

Type I (Figure 1A,B) was in 17 cases, 24 levels, Type II (Figure 2C,D) was in 28 cases, 38 levels, Type III (Figure 2E,F) was in 60 cases, 82 levels, and Type IV (Figure 3G,F) was in 75 cases, 106 levels. 

**Table 2 jcm-13-02666-t002:** Classification.

	Total	Type I	Type II	Type III	Type IV
	180 cases/250 levels	17 cases/24 levels	28 cases/38 levels	60 cases/82 levels	75 cases/106 levels
Sex male/female	105/75	10/7	17/11	34/26	45/30
Age (years)	25–83	25–80	29–75	29–77	26–83
	(average 45.6)	(average 42.2)	(average 38.7)	(average 43.2)	(average 44.8)
Follow-up period (years)	2.0–7.0	2.0–7.0	2.0–7.0	2.0–7.0	2.0–7.0
	(average 4.5)	(average 4.7)	(average 5.0)	(average 4.7)	(average 4.8)
1 level fusion	110 cases	10 cases	18 cases	38 cases	44 cases
2 level fusion	70 cases	7 cases	10 cases	22 cases	31 cases

### 3.2. Neurological Symptoms and Signs

Neurological symptoms improved in all cases. Deep axial pain at the neck and shoulders disappeared in 85% of cases and was significantly resolved in 15% of cases. Radicular pain in both upper extremities disappeared in 80% of cases and was significantly resolved in 20% of cases. Radiculopathy at both upper extremities disappeared in 75% of cases and was significantly resolved in 25% of cases. Myelopathy in both upper and lower extremities disappeared in 83% of cases and was significantly resolved in 17% of cases.

### 3.3. JOA Score and Its Improvement Rate (Table 3)

The overall JOA score IR was 78.4% at 6 months post-operatively, with the most recent evaluation showing an improvement of 77.5%. The JOA score IR in all types at 6 months post-operatively was ~75%, with the most recent evaluation showing an improvement of ~73%. The JOA score IR in Type I was 96.8% at 6 months post-operatively, with the most recent evaluation showing an improvement of 96.7%. The improvement was significantly higher in Type I than in the other groups (Type II, III, and IV). High amounts of improvements in neurological symptoms and activities of daily living were achieved in all types and maintained.

**Table 3 jcm-13-02666-t003:** JOA score and its improvement rate (I.R.).

Pre-op. JOA Score				Post-op. JOA Score		Post-op. I.R. (%) ± SD	
	Number	Range	Mean ± SD	Range	Mean ± SD	Range	Mean ± SD
				6 months after surgery			
Total	180	3.0–13.5	10.8 ± 1.7	15.0–17.0	15.7 ± 1.7	69.5–100	78.4 ± 2.5
				The latest			
				14.5–17.0	15.6 ± 1.67	65.0 -100	77.5 ± 1.7
				6 months after surgery			
Type I	17	12.0–14.5	13.5 ± 1.7 *	15.5–17.0	16.2 ± 1.5 *	92.5 -100	96.8 ± 2.3 *
				The latest			
				15.5–17.0	16.8 ± 1.7 *	90.2–100	96.7 ± 1.7 *
				6 months after surgery			
Type II	28	4.0–13.0	10.7 ± 1.5	15.0–17.0	15.8 ± 1.6	71.3–100	84.2 ± 2.5
				The latest			
				15.0–17.0	15.6 ± 1.8	68.4–100	78.1 ± 2.7
				6 months after surgery			
Type III	60	3.0–13.5	10.7 ± 1.5	15.0–17.0	15.5 ± 1.6		78.2 ± 2.5
				The latest			
				14.5–17.0	15.4 ± 1.7	66.8–100	75.1 ± 2.7
				6 months after surgery			
Type IV	75	3.0–13.5	10.6 ± 1.8	15.0–17.0	15.6 ± 1.9	62.4–100	75.2 ± 2.7
				The latest			
				14.5–17.0	15.5 ± 1.7	60.5–100	73.7 ± 2.8

Abbreviations: op. = operation, JOA = Japanese Orthopaedic Association (total points: 17.0). I.R. = improvement rate, SD = standard deviation. Significantly higher in Type I than in the other goups (Type II, III and IV). *: Significantly higher than that in the other types.

### 3.4. Neuroradiological Findings (Table 4 and Table 5) (Figure 8 and Figure 9)

There were no significant differences in the neuroradiological findings of the Type I cases post-operatively and the latest data compared with the pre-operative period. There was significantly larger C2-7 LA post-operatively compared to the pre-operative period in Type II, III, and IV. There was a significantly shorter C2-7 SVA post-operatively compared to the pre-operative period in Type II, III, and IV. There was a significantly smaller T1 slope post-operatively compared to the pre-operative period in Type II, III, and IV. In all types, a post-operative increase of lordosis was achieved and maintained, and also a deterioration of the anterior slope was avoided (Figure 10).

**Table 4 jcm-13-02666-t004:** C2-7 lordotic angle (LA), C2-7 sagittal vertical axis (SVA), and T1 slope.

	Number	C2-7 LA °				C2-7 SVA mm				T1 Slope °			
		Range	Mean ± SD	Range	Mean ± SD	Range	Mean ± SD	Range	Mean ± SD	Range	Mean ± SD	Range	Mean ± SD
		Pre-op.		post-op. 6 months		Pre-op.		post-op. 6 months			Pre-op.	post-op. 6 months	
Total	180	7.0–13.4	10.8 ± 1.7	12.0–17.5	15.5 ± 1.7 *	16.5–19.7	14.4 ± 2.5	10.5–18.0	16.0 ± 1.6	5.0–11.5	8.0 ±5.7	2.2–7.4	5.8 ± 1.8 #
				The latest				The latest				The latest	
				12.0–17.0	15.5 ± 1.7 *			12.4–18.5	16.3 ± 1.8			3.8–8.5	5.8 ± 1.7 #
				post-op. 6 months				post-op. 6 months				post-op. 6 months	
Type I	17	7.2–13.4	10.8 ± 1.7	12.0–17.5	12.1 ± 1.5	17.5–19.2	18.8 ± 2.3 *	12.7–17.8	18.0 ± 1.7	5.0–10.2	8.8 ± 5.7	1.0–7.0	5.0 ± 1.6
				The latest				The latest				The latest	
				12.0–17.0	11.8 ± 1.4			12.9–18.2	17.8 ± 1.5			3.8–7.2	5.5 ± 1.5
				post-op. 6 months				post-op. 6 months				post-op. 6 months	
Type II	25	7.3–13.2	10.7 ± 1.5	12.0–17.3	15.8 ± 1.6 *	16.5–19.5	17.2 ± 2.5 *	10.5–17.5	15.2 ± 1.6 **	5.0–11.4	8.6 ± 5.7	2.2–7.0	5.0 ± 1.9 #
				The latest				The latest				The latest	
				12.0–16.5	15.7 ± 1.4 *			12.4–18.5	15.8 ± 1.8 **			3.5–6.5	5.8 ± 1.5 #
				post-op. 6 months				post-op. 6 months				post-op. 6 months	
Typer III	58	7.5–13.0	10.7 ± 1.5	12.2–17.5	16.0 ± 1.6 *	16.5–19.5	17.5 ± 2.5	12.6–18.0	15.8 ± 1.6 **	5.4–11.1	8.0 ± 6.0	4.0–7.0	6.0 ± 1.7 #
				The latest				The latest				The latest	
				12.0 16.3	15.7 ± 1.4 *			12.9–18.5	16.2 ± 1.8 **			4.5–8.5	6.2 ± 1.8 #
				post-op. 6 months				post-op. 6 months				post-op. 6 months	
Type IV	80	7.0–13.0	10.5 ± 1.8	12.0–17.2	16.2 ± 1.9 *	16.9–19.7	19.2 ± 2.7	12.8–17.8	15.5 ± 1.7 **	5.0–11.5	8.7 ± 5.8	4.4–7.4	4.5 ± 1.5 #
				The latest				The latest				The latest	
				12.0 17.0	15.7 ± 1.4 *			13.2–18.2	16.2 ± 1.8 **			5.0–8.2	5.5 ± 1.5 #

Abbreviations: C = cervical spine, LA = lordotic angle, SVA = sagittal vertical axis, T1 = the first thoracic spine, SD = standard deviation, op. = operation. °: degrees. *: Significantly larger in post-operatively and in the latest data than in the pre-operative period. **: Significantly shorter in post-operatively and in the latest data than in the pre-operative period. #: Significantly smaller in post-operatively and in the latest data than in the pre-operative period.

In all types, the mean IVS increased post-operatively and the distance was maintained at the latest assessment. There was no significant difference in the IVS between all Types both pre- and post-operatively. A post-operative increase in the IVS was achieved and maintained. The data demonstrated that the alignment can be reduced, and good alignment can be maintained while avoiding subsidence (Figure 10).

**Table 5 jcm-13-02666-t005:** Intervertebral space (IVS).

		Pre-op. Mean IVS		Post-op. 6 Months Mean IVS		The Latest Mean IVS	
		Range (mm)	Mean ± SD (mm)	Range (mm)	Mean ± SD (mm)	Range (mm)	Mean ± SD (mm)
Total	180	1.5–7.0	3.7 ± 1.7	4.5–6.8	5.8 ± 1.7 *	4.3–6.6	5.5 ± 0.4
Type I	17	2.0–6.5	4.4 ± 1.9	4.8–6.8	6.2 ± 0.7 *	4.7–6.6	6.0 ± 0.4 *
Type II	25	1.5–6.8	3.9 ± 1.2	4.8–6.5	5.5 ± 0.6 *	4.7–6.3	5.3 ± 0.3 *
Typer III	58	1.5–7.0	3.5 ± 1.8	4.7–6.5	5.8 ± 0.7 *	4.5–6.4	5.5 ± 2.5 *
Type IV	80	1.0–6.9	3.6 ± 1.8	4.5–6.8	6.0 ± 0.7 *	4.3–6.6	5.8 ± 2.7 *

Abbreviations: IVS = intervertebral space, SD = standard deviation, op. = operation. *: Significantly larger in post-operatively and in the latest data than in the pre-operative period.

**Figure 10 jcm-13-02666-f010:**
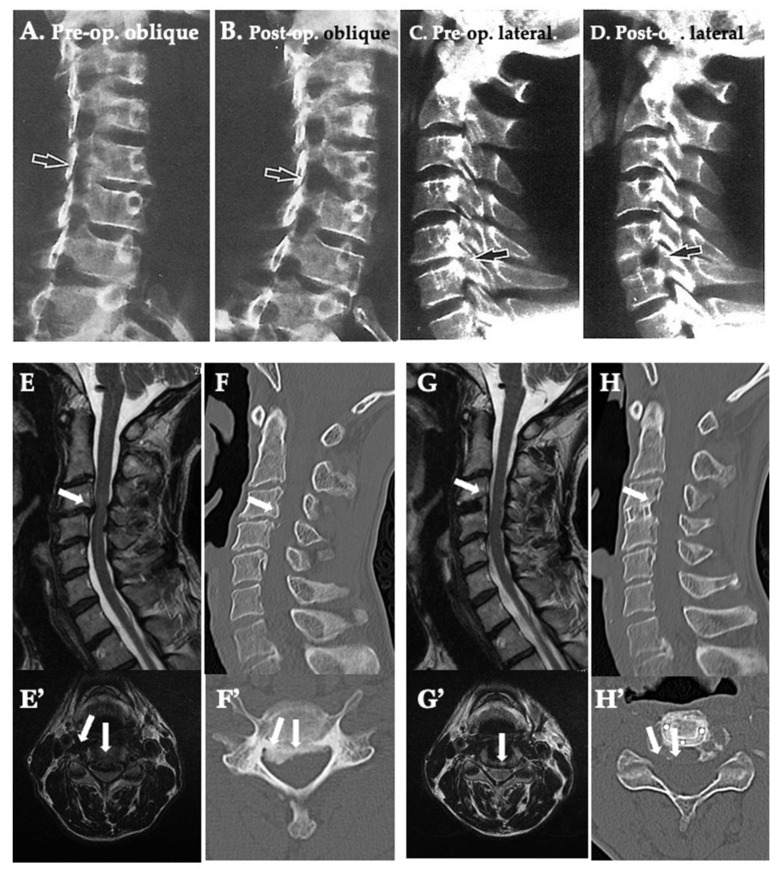
Pre- and Post-operative plain X-ray, two dimensional computed tomography (2D-CT), and magnetic resonance (MR) images in the neutral position. (**A**) Pre-operative (op.) cervical spine oblique plain X-ray demonstrating the osteophyte at the intervertebral foramen (IVF) of the right C5/6 (black arrow). (**B**) Post-op. cervical spine oblique plain X-ray demonstrating an opened IVF by the perfect resection of the osteophytes (black arrow). (**C**) Pre-op. cervical spine lateral plain X-ray demonstrating the osteophyte at IVF of the dorsal C5/6 (black arrow). (**D**) Post-op. cervical spine lateral plain X-ray demonstrating an opened IVF by perfect resection of osteophytes (black arrow). (**E**,**E’**) Pre-op. T2 weighted MR sagittal and axial images demonstrating a low-intensity mass compressing the cord at the C3/4 level (white arrows). (**F**,**F’**) Pre-op. 2D-CT sagittal and axial images demonstrating the osteophyte in the median and IVF at C3/4 level (white arrows). (**G**,**G’**) Post-op. T2 weighted MR sagittal and axial images demonstrating the removed low-intensity mass and resolution of the compression of the cord (white arrows). (**H**,**H’**) Post-op. 2D-CT sagittal and axial images demonstrate the removal of the osteophyte (white arrows) and the cage in the appropriate position.

### 3.5. Complications (Table 6)

There was no mortality and transient morbidity was observed in five cases (2.7% overall). Swallowing disturbance and hoarseness occurred in three cases (1.7%): one type III case (2.9%) and two type IV cases (4.4%). The swallowing disturbance and hoarseness were caused by swelling of the vocal cord and esophagus due to traction and compression, not due to injury of recurrent laryngeal nerve. The absence of movement disturbance of the vocal cord was confirmed by fiberoptic examination. These symptoms resolved in several days. C5 palsy occurred in one type III case (2.9%), which was resolved in a couple of months by rehabilitation. Horner’s syndrome occurred in one type IV case (2.2%) but it resolved in a couple of months.

**Table 6 jcm-13-02666-t006:** Complications.

	Total	Type I	Type II	Type III	Type IV
	5 cases (2.7%)	0	0	2 (3.3%)	3 (4.0%)
Swallowing disturbance and Hoarsness					
	3 cases (1.7%)	0	0	1 (2.9%)	2 (4.4%)
C5 palsy	0	0	0	1 (2.9%)	
Hornel’s syndrome	0	0	0	0	1 (2.2%)

## 4. Discussion

### 4.1. Review of Anterior Cervical Decompression (ACD) and the Introduction of the TUD Approach

Surgery for cervical spondylosis and discogenic disease is generally divided into two approaches; the posterior and the anterior approach. Robinson and Smith [12] in 1955 and Cloward [13] in 1958 pioneered the anterior cervical discectomy with bone fusion to accomplish direct decompression of the compressive spondylotic spur and disc fragment. In 1988, Whitecloud [14] pointed out the following problems and limitations of the surgical procedures introduced by Cloward, Smith, and Robbinson [12,13,14]: (1) insufficient resection of lateral osteophyte, as it is difficult to open the intervertebral foramen; (2) vertical direction for manipulation of the osteophyte and lesion.

In 1976, Hakuba [1] introduced the TUD approach, which is a combined anterior and lateral approach to cervical discs. In addition to resecting the uncovertebral joint, the entire disc is removed including the ipsilateral posterior osteophyte and the contralateral uncinate process. Snyder and Bernhardt [3], in 1989, reported an anterior cervical fractional interspace decompression for the treatment of cervical radiculopathy. Decompression is for the vertebral disc and is limited to the lateral one-third of the intervertebral disc and is limited to a radius around the nerve root. Joh [4] in 1996 described a modified approach that completely exposes the vertebral artery when the entire uncinate process is removed. Lee [5] in 2006 described a small keyhole transuncal foraminotomy for unilateral cervical radiculopathy with preservation of the intervertebral disc. The TUD described in the present study is greatly different from their previous methods. In the TUD approach under direct vision using an operative microscope, we can safely and completely remove extensive lateral spurs without exposing the vertebral artery in its canal. Extensive removal of the posterolateral corner and transverse ridge of the vertebral body allows sufficient space for the nerve root and spinal cord, there is little danger of encroachment of the intervertebral foramen and spinal canal.

### 4.2. Application of TUD Approach

We divided the TUD approach into three procedures: (1) unco-foraminotmy; (2) removal of the lesion and resection of the dorsal osteophyte of the vertebral body; (3) unco-foraminotomy at the contralateral side. Furthermore, we classified the cases into four types based on the location of compression of the nerve roots and the spinal cord.

Type I: compression of the nerve roots in the intervertebral foramen. This type is indicated above (1) unco-foraminotomy.Type II: compression of the spinal cord in the posterior direction due to disc hernia, osteophytes, and ossification of the posterior longitudinal ligament. This type is indicated above as (2) removal of the lesion and resection of the dorsal osteophyte of the vertebral body.Type III: compression of nerve roots and the spinal cord occurs posteriorly from the vertebral foramen. This type is indicated above (1) and (2).Type IV: in addition to Type III, compression of the spinal cord and nerve roots on the opposite side of the intervertebral foramen. This type is indicated above (1), (2), and (3) unco-foraminotomy at the contralateral side.

By applying the three procedures in the TUD approach appropriately to these four types, it is possible to remove lesions effectively. The surgery can be performed with minimal invasiveness tailored to the specific pathology, allowing for optimal outcomes and reducing the occurrence of complications.

### 4.3. Surgical Outcome, Factors Leading Good Outcome Sand Comparison with Other Reports

We applied the TUD approach based on the location of the compression, demonstrating its utility, our modifications, and considerations. The TUD approach allows for direct visualization and safe, reliable lesion removal of lesions in a less invasive manner than the previous approach. It is highly effective and safe, resulting in a high improvement rate of neurological symptoms and activity of daily living, and minimal complication (Table 3). By opening the intervertebral foramen and Luschka’s joint, and setting of cages, the alignment can be reduced and good alignment can be achieved and maintained while avoiding subsidence. Anterior cervical decompression (ACD) by TUD achieved and maintained good neuroradiological alignment and sagittal balance (Table 4 and Table 5).

In this study, the outcome was better than those in previous reports of ACD [15,16,17,18,19,20,21,22]. Several authors reported the efficacy of the various anterior foraminotomy. Matz [16] reviewed the indications and utility of anterior cervical nerve root decompression. In summary, the success rates were 52–99% but recurrent symptoms were as high as 30% [15,16,17,18,19,20,21,22]. Conversely, the efficacy of posterior cervical foraminotomy for cervical radiculopathy has been reported, and in a review [23,24,25], McAnany et al. [26] stated that there was no significant difference in the pooled outcome between a traditional open or minimally invasive foraminotomy using a tubular retractor. The pooled clinical success rate was 92.7% for open foraminotomy and 94.9% for minimally invasive foraminotomy. But the improvement rate in these reports was not better than that in this report. We guess that the foraminotomy in these reports is not enough and not appropriate [15,16,17,18,19,20,21,22,23,24,25,26].

### 4.4. Comparison between Advantages of the TUD Approach and Disadvantages of the Other Procedures

(1)Advantages of the TUD approach (Figure 3, Figure 4, Figure 5 and Figure 6):

We use a vertebral spreader and manually operate a retractor. By doing so, tools such as a drill, curette, Kerrison rongeur, and bipolar forceps can be inserted diagonally into the surgical field, utilizing the space between the legs of the spreader. This allows the surgical field and manipulation site to be operated on while they are visualized directly, as the line of sight and these tools do not overlap [1,2,6,7]. Consequently, surgery can be performed safely without damaging the dura mater, venous plexus, vertebral artery, and nerve roots, enabling the reliable removal of lesions [1,2,6,7].

(1’)Disadvantages of the other procedures:

Expanding up to the vertebral body, many surgical techniques are used to secure the surgical field, with surgeons utilizing fixed retractors [12,13,14,15,16]. However, only the surgical field within the range of the retractor can be obtained and it is not possible to secure adequately the surgical field outside the retractor. Furthermore, tools such as forceps, suction, bipolar forceps, and drills cannot be applied outside of this surgical field. During surgery, procedures are limited to the area under direct vision. Therefore, there are limitations regarding the removal of the outer uncinate process, decompression of the intervertebral foramen, and release of the Luschka’s joint [12,13,14,15,16].

(2)Advantages of the TUD approach (Figure 3, Figure 4 and Figure 6):

Unco-foraminotomy allows for the removal of the uncinate process, securing a working space for surgical operations in that area. By opening the intervertebral foramen and Luschka’s joint and setting of cages, the alignment can be reduced and good alignment can be achieved.

(2’)Disadvatage of the other procedures:

In conventional surgical methods, it is not possible to open the outer intervertebral foramen. Attempting to open the intervertebral foramen with the same approach would involve obliquely transversing the intervertebral space and necessitate the complete removal of the intervertebral disc, which results in significant invasiveness in certain cases such as Type I [12,13,14,15,16].

While the transcorporeal approach offers the advantage of direct access to a compressive lesion from the anterior direction, this increases the use of the vertebral body as supporting tissue, thereby increasing the long-term risk of the vertebral body compression [27,28].

While the posterior approach removes compressive lesions such as intervertebral disc herniation and osteophytes located anteriorly to the nerve root, it becomes necessary to operate beyond the nerve root, inevitably leading to some degree of nerve root retraction. This increases the risk of nerve root damage [29,30,31].

(3)Advantages of the TUD approach

By taking an oblique view, traction on the pharynx, larynx, esophagus, and nerves surrounding the vertebral and carotid arteries can be minimized. Additionally, we alternate between a vertebral spreader and a manual retractor every 15–20 min instead of using a fixed retractor. This prevents prolonged retraction of the pharynx, larynx, esophagus, and nerves in the surrounding area. As a result, complications such as C5 palsy, swallowing disturbance, and hoarseness due to soft tissue damage and swelling, including the pharynx, larynx, and esophagus, and manifestations of Horner’s syndrome due to disorders of the sympathetic nerve trunk are believed to be less likely to occur (Table 6) [15,16,17,18,19,20,21,22].

(3’)Disadvantages of the other procedures:

In other surgical methods, fixed retractors are used to secure and maintain the surgical field, but this can exert strong retraction on the esophagus and pharynx, potentially leading to the aforementioned complications.

### 4.5. Caring Points in Technical Aspects and Limitations in the Application of the TUD Approach

This ensures that disc removal/excision is limited to approximately the outer one-third, eliminating the need for fixation in Type I and Type IV cases and preventing post-operating instability [1].

Since the surgical field is observed from an oblique angle, it is crucial to proceed with the operation while confirming the exact center. The manual retractor is manipulated and held by an assistant. However, it is necessary to accurately pull and protect the esophagus, pharynx, larynx, and carotid sheath.

It is important to minimize traction in the esophagus, pharynx, and larynx during cage insertion, as is required for inserting cages straight from the midline, similar to artificial intervertebral discs, aiming to minimize traction as much as possible and for a short duration during the insertion procedure (Figure 7).

The TUD approach can be applied between C2/3 and C7/Th1 according to anatomical limitations. In the upper limitation, the mandibular bone hinders its use, while in the caudal limitation, the subauricular bone prevents its use. It is difficult to apply the TUD approach to lesions over the spinal cord and nerve roots.

### 4.6. Modified and Advanced Procedures

We have extended the application of the TUD approach to include vertebral body resection, addressing lesions not only in the posterior aspect of the vertebral body such as ossification of the posterior longitudinal ligament, but also lesions inside and outside the dura mater on the ventral side of the spinal cord. We have applied the TUD approach to lesions both inside and outside the spinal cord [32,33,34,35].

We have also applied the TUD approach to antero-lateral vertebrectomy [35]. Various methods of anterior foraminotomy and transcorporeal anterior cervical foraminotomy have been proposed [36,37,38,39,40]. More recently anterior transcorporeal procedure using computed tomography-based intraoperative spinal navigation and percutaneous endoscopy was reported [36,39]. In addition, their combination has been described [38,40]. As a new modality, endoscopy has been applied [41]. The TUD approach can be combined with these new procedures.

## 5. Conclusions

We classified our cases into four groups based on the location of compressive lesions in the spinal cord and/or roots. We modified and applied the TUD approach for each group. We presented the surgical techniques for the modified TUD approach for each group. We suggested the advantage and utility of the TUD approach as a surgical procedure for cervical spondylotic myelopathy and radiculopathy, incorporating innovations.

The TUD approach allows the direct visualization and safe, reliable removal of the lesion. It is highly effective and safe, resulting in a high improvement rate of neurological symptoms and minimal complications. Anterior discectomy and fusion by TUD achieved and kept good alignment and sagittal balance.

The TUD approach is highly versatile and can be applied to vertebrectomy. If the TUD approach and endoscopic approaches can be combined, their application to new area is anticipated.

## Data Availability

All data are available on request to the corresponding author. The data are not publicly available due to privacy.

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
