# Peer review of "Application and Advantages of the Trans-Unco-Discal (TUD) Approach for Cervical Spondylotic Myelopathy and Radiculopathy: Classification and Modification of Surgical Technique Based on the Location of Spinal Cord and/or Nerve Root Compression"

_jcm, 2024, doi:10.3390/jcm13092666_

Round 1

Reviewer 1 Report

Comments and Suggestions for Authors

The authors present their results with a variation of the ACDF technique, as well as their classification of cord / nerve compression. The results are reasonable, but the major problem is that ACDF (including their technique variation) is literally the most common procedure done by spine surgeons. Moreover, there is no need for a "classification" of cord / nerve compression; we just note where the compression is, and make sure that area is decompresed.

Comments on the Quality of English Language

English can be significantly improved. 

Author Response

Moriguchi-Ikuno Memorial Hospital, Koudoukai Health System

Department of Neurosurgery

Craniocervical Junction and Spine Center

Lindsay A. Farrer, M.D. Editor-in-Chief

Massimiliano Visocchi, M.D. Special Issue Editor

Holy Guo. Section Managing Editor

Journal of Clinical Medicine, Editorial Office MDPI, Basel, Switzerland

Dear Drs.

I deeply appreciate to the reviewers and the editors for checking our article at detail and gave us the kindly advices and suggestions.

I am pleasure to submit the revised manuscript entitled, I.D. 2922319: “Application and Advantages of Trans-Unco-Discal (TUD) Approach for Cervical Spondylotic Myelopathy and Radiculopathy: classification and modification of surgical technique based on the location of spinal cord and/or nerve root compression”. I revised according to the suggestions and advises of the reviewers. This manuscript was edited by the scientific English native speaker. I attached its certification in the next page.

As corresponding author, I look forward to dealing with the paper prepublication.

Sincerely yours.

Misao Nishikawa, M.D.Ph.D.

Department of Neurosurgery

Moriguchi-Ikuno Memorial Hospital, Koudoukai Health System

6-17-33 Satanakamachi, Moriguchi City, 570-0002 Osaka Japan

Phone: +81-6-6906-1100    Fax: +81-6-6902-9070  

Moriguchi-Ikuno Memorial Hospital, Koudoukai Health System

Department of Neurosurgery

Craniocervical junction and Spine Center

Journal of Clinical Medicine, MDPI Journals Department “Application and Advantages of Trans-Unco-Discal (TUD) Approach for Cervical Spondylotic Myelopathy and Radiculopathy: classification and modification of surgical technique based on the location of spinal cord and/or nerve root compression”

In consideration of the editor’s and publisher’s expense and effort in reviewing, editing and publishing the Manuscript, and of the professional benefits related to its publication, I hereby transfer, assign and otherwise convey to MDPI, Journal of Clinical Medicine upon acceptance of the Manuscript by Journal of Clinical Medicine, all rights and interest in the Manuscript, including, copyright ownership, together with full right and authority to the manuscript in all forms and media and claim worldwide copyright for that published.

Each of undersigned is author of the Manuscript and all authors are named on this document.

All intellectual contributions, technical help, financial or material support, and all financial or other relations that may contribute or lead to a conflict of interest have been acknowledged or disclosed in the Manuscript.

The authors report no conflict of interest relevant to research. All authors are members of the Japanese Neurological Society (JNS) have registered online and filled the self-reported COI disclosure statement form on the JNS website.

This report has been proven by the Institutional Review Board of Moriguchi-Ikuno Memorial Hospital (2018-1-001) and Osaka Metropolitan University Graduate School of Medicine (2019-2-4234).

The Manuscript is not subject to copyright or rights except for my own to be transferred upon public.

Manuscript ID: jcm-2922319

Dear Reviewer 1

We deeply sincerely appreciate your review of our manuscript. The manuscript has been checked by a native English speaker.

We have revised the manuscript according to your suggestions as below. Please see my responses below.

We have the red highlight where we added and revised text.

1: We have revised the Introduction, and Discussion, according to the pointed focus by the reviewer 1, and the red highlighted the revised text.

Anterior cervical decompression (ACD) and anterior cervical descectomy and fusion (ACDF) are common procedures in the field of spinal surgery, but the detail surgical methods vary. Currently, the assessment of compression sites on the spinal cord and nerve roots, as well as the degree of decompression achieved, is left to discretion of the individual surgeons. I think that it is important to perform more appropriate decompression and less invasive surgery. Therefor it is necessary to classify compression sites to the spinal cord and/or nerve roots in order to accurately identify the areas that require bone removal and decompression pre-operatively, as well as to assess the degree and achievement of decompression. The TUD approach we have been using is a surgical approach that can address various pathology compression sites. Furthermore, TUD approach offers a safer procedure to confirm the degree and achievement of decompression.

Reviewer 2 Report

Comments and Suggestions for Authors

The manuscript reported in detail for modified AUD procedures and specified surgical tools. They also reported different approaches operating on different subtypes of cervical compression. The classification is a clinically usefully neuroradiological scaling to discriminate the severity. Overall, I think this is a very valuable study but I have below questions:

1.     Line 49, misspelling ‘usefulness’.

2.     Lines 290, 291, misspelling ‘improvement’.

3.     2.6.1 Please briefly describe in detail the JOA scoring methods.

4.     Did the surgical procedures specifically pay attention on recurrent laryngeal nerve protection?

5.     Any quantitative physical exams or EMG done for pain, numbness, or motor functions?

Comments on the Quality of English Language

There are many spelling and grammar mistakes. Though they did not make it very hard to follow, but please put significant effort on this.

Author Response

Moriguchi-Ikuno Memorial Hospital, Koudoukai Health System

Department of Neurosurgery

Craniocervical Junction and Spine Center

Lindsay A. Farrer, M.D. Editor-in-Chief

Massimiliano Visocchi, M.D. Special Issue Editor

Holy Guo. Section Managing Editor

Journal of Clinical Medicine, Editorial Office MDPI, Basel, Switzerland

Dear Drs.

I deeply appreciate to the reviewers and the editors for checking our article at detail and gave us the kindly advices and suggestions.

I am pleasure to submit the revised manuscript entitled, I.D. 2922319: “Application and Advantages of Trans-Unco-Discal (TUD) Approach for Cervical Spondylotic Myelopathy and Radiculopathy: classification and modification of surgical technique based on the location of spinal cord and/or nerve root compression”. I revised according to the suggestions and advises of the reviewers. This manuscript was edited by the scientific English native speaker. I attached its certification in the next page.

As corresponding author, I look forward to dealing with the paper prepublication.

Sincerely yours.

Misao Nishikawa, M.D.Ph.D.

Department of Neurosurgery

Moriguchi-Ikuno Memorial Hospital, Koudoukai Health System

6-17-33 Satanakamachi, Moriguchi City, 570-0002 Osaka Japan

Phone: +81-6-6906-1100    Fax: +81-6-6902-9070  

Moriguchi-Ikuno Memorial Hospital, Koudoukai Health System

Department of Neurosurgery

Craniocervical junction and Spine Center

Journal of Clinical Medicine, MDPI Journals Department “Application and Advantages of Trans-Unco-Discal (TUD) Approach for Cervical Spondylotic Myelopathy and Radiculopathy: classification and modification of surgical technique based on the location of spinal cord and/or nerve root compression”

In consideration of the editor’s and publisher’s expense and effort in reviewing, editing and publishing the Manuscript, and of the professional benefits related to its publication, I hereby transfer, assign and otherwise convey to MDPI, Journal of Clinical Medicine upon acceptance of the Manuscript by Journal of Clinical Medicine, all rights and interest in the Manuscript, including, copyright ownership, together with full right and authority to the manuscript in all forms and media and claim worldwide copyright for that published.

Each of undersigned is author of the Manuscript and all authors are named on this document.

All intellectual contributions, technical help, financial or material support, and all financial or other relations that may contribute or lead to a conflict of interest have been acknowledged or disclosed in the Manuscript.

The authors report no conflict of interest relevant to research. All authors are members of the Japanese Neurological Society (JNS) have registered online and filled the self-reported COI disclosure statement form on the JNS website.

This report has been proven by the Institutional Review Board of Moriguchi-Ikuno Memorial Hospital (2018-1-001) and Osaka Metropolitan University Graduate School of Medicine (2019-2-4234).

The Manuscript is not subject to copyright or rights except for my own to be transferred upon public.

Manuscript ID: jcm-2922319

Dear Reviewers 2

We deeply sincerely appreciate your review of our manuscript. The manuscript has been checked by a native English speaker.

We have revised the manuscript according to your suggestions as below. Please see my responses below.

We have the red highlight where we added and revised text.

1,2: I revised the spelling.

3: I added the table (Table 1) to describe in detail the JOA scoring methods, according to the suggestion of reviewer 2.

4: In the case of the left-sided approach, the recurrent laryngeal nerve runs along the back of the esophagus, and care is taken not to detach the posterior aspect of the esophagus while reaching the anterior aspect of the vertebral body. Additionaly, when exposing the esophagus along of the vertebral body, slightly traction of the esophagus along with the deep cervical fascia is performed. On the other hand, in the approach from the right side, the recurrent laryngeal nerve runs along the side of the esophagus, so efforts are made to minimize traction on the esophagus more conservatively. Sometimes, operations are performed while protection with a thin cotton sheath.

           This information has been added to 2.4. Surgical procedure of the original ventral vertebral body, and burr hole (Figure 3A-3D).

  1. Electromyography (EMG) and somatosensory evoked potentials (SEP) are not performed in all cases pre-and post-operatively. It depends on the case. However, intraoperative monitoring of EG and SEP is considered essential. We have not addressed this in the current paper as it is not the focus of this paper. We intend to present this in future opportunities.

Reviewer 3 Report

Comments and Suggestions for Authors

Dear Author:

As an invited reviewer for this journal, I have reviewed your paper. Overall, this is a valuable technical article that extensively discusses the modified trans-unco-discal (TUD) surgical technique and its indications, while categorizing various types of lesions and discussing surgical approaches. My main suggestions are:

Firstly, Radiological assessment indicators can include grading of disc degeneration, changes in intervertebral space height, and other measures to evaluate the impact of TUD approach on the intervertebral disc.

Secondly, The discussion section could incorporate a comparison of the advantages and disadvantages with other traditional surgical approaches.

I hope these constructive comments can help improve your paper. 

Comments on the Quality of English Language

Dear Author,

Regarding the English language quality of this manuscript, I have the following comments:

In general, the language of this paper is fluent and smooth. The use of vocabulary and terminology is accurate. However, there are still some areas that need improvement:

There are some minor errors in grammar and word usage that need to be corrected. In some places, the expressions seem too rigid or verbose and could be improved to be more concise and fluent.

I hope these comments can be helpful for language editing of this manuscript. Please feel free to contact me if you have any other questions.

Author Response

Moriguchi-Ikuno Memorial Hospital, Koudoukai Health System

Department of Neurosurgery

Craniocervical Junction and Spine Center

Lindsay A. Farrer, M.D. Editor-in-Chief

Massimiliano Visocchi, M.D. Special Issue Editor

Holy Guo. Section Managing Editor

Journal of Clinical Medicine, Editorial Office MDPI, Basel, Switzerland

Dear Drs.

I deeply appreciate to the reviewers and the editors for checking our article at detail and gave us the kindly advices and suggestions.

I am pleasure to submit the revised manuscript entitled, I.D. 2922319: “Application and Advantages of Trans-Unco-Discal (TUD) Approach for Cervical Spondylotic Myelopathy and Radiculopathy: classification and modification of surgical technique based on the location of spinal cord and/or nerve root compression”. I revised according to the suggestions and advises of the reviewers. This manuscript was edited by the scientific English native speaker. I attached its certification in the next page.

As corresponding author, I look forward to dealing with the paper prepublication.

Sincerely yours.

Misao Nishikawa, M.D.Ph.D.

Department of Neurosurgery

Moriguchi-Ikuno Memorial Hospital, Koudoukai Health System

6-17-33 Satanakamachi, Moriguchi City, 570-0002 Osaka Japan

Phone: +81-6-6906-1100    Fax: +81-6-6902-9070  

Moriguchi-Ikuno Memorial Hospital, Koudoukai Health System

Department of Neurosurgery

Craniocervical junction and Spine Center

Journal of Clinical Medicine, MDPI Journals Department “Application and Advantages of Trans-Unco-Discal (TUD) Approach for Cervical Spondylotic Myelopathy and Radiculopathy: classification and modification of surgical technique based on the location of spinal cord and/or nerve root compression”

In consideration of the editor’s and publisher’s expense and effort in reviewing, editing and publishing the Manuscript, and of the professional benefits related to its publication, I hereby transfer, assign and otherwise convey to MDPI, Journal of Clinical Medicine upon acceptance of the Manuscript by Journal of Clinical Medicine, all rights and interest in the Manuscript, including, copyright ownership, together with full right and authority to the manuscript in all forms and media and claim worldwide copyright for that published.

Each of undersigned is author of the Manuscript and all authors are named on this document.

All intellectual contributions, technical help, financial or material support, and all financial or other relations that may contribute or lead to a conflict of interest have been acknowledged or disclosed in the Manuscript.

The authors report no conflict of interest relevant to research. All authors are members of the Japanese Neurological Society (JNS) have registered online and filled the self-reported COI disclosure statement form on the JNS website.

This report has been proven by the Institutional Review Board of Moriguchi-Ikuno Memorial Hospital (2018-1-001) and Osaka Metropolitan University Graduate School of Medicine (2019-2-4234).

The Manuscript is not subject to copyright or rights except for my own to be transferred upon public.

Manuscript ID: jcm-2922319

Dear Reviewer 3

We deeply sincerely appreciate your review of our manuscript. The manuscript has been checked by a native English speaker.

We have revised the manuscript according to your suggestions as below. Please see my responses below.

We have the red highlight where we added and revised text.

1: Data measuring the intervertebral space has been added to Table 5., and the method and results have been added to appropriate sections. Furthermore, discussion of the implications of these results has been added to Discussion 4.3. Surgical outcome, factors leading to good results.

  1. In discussion, I added comparison between advantages of the TUD approach and disadvantages of the other procedures, and added eight papers (references: 23, 24, 25 (surgical outcome), 27, 28, 29, 30, 31) as the other procedures and cited them in each part.

Reviewer 4 Report

Comments and Suggestions for Authors

The authors are trying to assess the application and advantages of modifying the trans-unco-discal (TUD) approach which underwent for cervical myelo-radiculopathy. In the cohort they present the surgical techniques of the modified TUD approach. Total group consist of 180 cases who were performed anterior cervical decompression (ACD) by the modified TUD approach. All cases classified into four groups based on the location of nerve root and/or spinal cord compression. The authors concluded that ACD by the modified TUD approach safely achieved appropriate decompression for the spinal cord and/or nerve roots, and the patients had a high improvement rate and good alignment. Complications were less common than with other surgical procedures. 

I congratulate the authors on their solid and laborious work. I would like to point out a few details that could have improved the article. I would suggest adding a more extensive explanation in the introduction regarding the analyzed surgical techniques. The methodology is very well done. I would show the complications in a chart. The discussion is clear and so are the conclusions.

Author Response

Moriguchi-Ikuno Memorial Hospital, Koudoukai Health System

Department of Neurosurgery

Craniocervical Junction and Spine Center

Lindsay A. Farrer, M.D. Editor-in-Chief

Massimiliano Visocchi, M.D. Special Issue Editor

Holy Guo. Section Managing Editor

Journal of Clinical Medicine, Editorial Office MDPI, Basel, Switzerland

Dear Drs.

I deeply appreciate to the reviewers and the editors for checking our article at detail and gave us the kindly advices and suggestions.

I am pleasure to submit the revised manuscript entitled, I.D. 2922319: “Application and Advantages of Trans-Unco-Discal (TUD) Approach for Cervical Spondylotic Myelopathy and Radiculopathy: classification and modification of surgical technique based on the location of spinal cord and/or nerve root compression”. I revised according to the suggestions and advises of the reviewers. This manuscript was edited by the scientific English native speaker. I attached its certification in the next page.

As corresponding author, I look forward to dealing with the paper prepublication.

Sincerely yours.

Misao Nishikawa, M.D.Ph.D.

Department of Neurosurgery

Moriguchi-Ikuno Memorial Hospital, Koudoukai Health System

6-17-33 Satanakamachi, Moriguchi City, 570-0002 Osaka Japan

Phone: +81-6-6906-1100    Fax: +81-6-6902-9070  

Moriguchi-Ikuno Memorial Hospital, Koudoukai Health System

Department of Neurosurgery

Craniocervical junction and Spine Center

Journal of Clinical Medicine, MDPI Journals Department “Application and Advantages of Trans-Unco-Discal (TUD) Approach for Cervical Spondylotic Myelopathy and Radiculopathy: classification and modification of surgical technique based on the location of spinal cord and/or nerve root compression”

In consideration of the editor’s and publisher’s expense and effort in reviewing, editing and publishing the Manuscript, and of the professional benefits related to its publication, I hereby transfer, assign and otherwise convey to MDPI, Journal of Clinical Medicine upon acceptance of the Manuscript by Journal of Clinical Medicine, all rights and interest in the Manuscript, including, copyright ownership, together with full right and authority to the manuscript in all forms and media and claim worldwide copyright for that published.

Each of undersigned is author of the Manuscript and all authors are named on this document.

All intellectual contributions, technical help, financial or material support, and all financial or other relations that may contribute or lead to a conflict of interest have been acknowledged or disclosed in the Manuscript.

The authors report no conflict of interest relevant to research. All authors are members of the Japanese Neurological Society (JNS) have registered online and filled the self-reported COI disclosure statement form on the JNS website.

This report has been proven by the Institutional Review Board of Moriguchi-Ikuno Memorial Hospital (2018-1-001) and Osaka Metropolitan University Graduate School of Medicine (2019-2-4234).

The Manuscript is not subject to copyright or rights except for my own to be transferred upon public.

Manuscript ID: jcm-2922319

Dear Reviewer 4

We deeply sincerely appreciate your review of our manuscript. The manuscript has been checked by a native English speaker.

We have revised the manuscript according to your suggestions as below. Please see my responses below.

We have the red highlight where we added and revised text.

1: I added extensive planation about the techniques of modified TUD approach within the descriptions of each operation in the first paragraph of Introduction.

In TUD approach under direct vision using the operative microscope, we have been able to remove safely and completely extensive lateral spurs without exposing the vertebral artery in its canal. There is less risk of injury to the spinal cord in TUD approach than there is by a transdiscal approach. Extensive removal of the posterolateral corner and transverse ridge of the vertebral body allows sufficient space for the nerve roots and spinal cord, and there is little danger of encroachment of the intervertebral foramen and spinal canal.

  1. I added a table of complications added as Table 6.

Round 2

Reviewer 1 Report

Comments and Suggestions for Authors

no new comments

Author Response

Manuscript ID: jcm-2922319

Dear Reviewer 1

We deeply sincerely appreciate your review of our manuscript. The manuscript has been checked by a native English speaker.

           In Round 2, you have no new comment.

We have revised the manuscript according to your suggestions in Round 1 as below. Please see my responses below.

We have the red highlight where we added and revised text.

1: We have revised the Introduction, and Discussion, according to the pointed focus by the reviewer 1, and the red highlighted the revised text.

Anterior cervical decompression (ACD) and anterior cervical descectomy and fusion (ACDF) are common procedures in the field of spinal surgery, but the detail surgical methods vary. Currently, the assessment of compression sites on the spinal cord and nerve roots, as well as the degree of decompression achieved, is left to discretion of the individual surgeons. I think that it is important to perform more appropriate decompression and less invasive surgery. Therefor it is necessary to classify compression sites to the spinal cord and/or nerve roots in order to accurately identify the areas that require bone removal and decompression pre-operatively, as well as to assess the degree and achievement of decompression. The TUD approach we have been using is a surgical approach that can address various pathology compression sites. Furthermore, TUD approach offers a safer procedure to confirm the degree and achievement of decompression.